# Detecting underreporters of abortions and miscarriages in the national study of family growth, 2011–2015

**Ting Yan** *, **Roger Tourangeau**

Westat, Rockville, Maryland, United States of America

* tingyan@westat.com

## Abstract

This paper draws on individual-level data from the National Study of Family Growth (NSFG) to identify likely underreporters of abortion and miscarriage and examine their characteristics. The NSFG asks about abortion and miscarriage twice, once in the computer-assisted personal interviewing (CAPI) part of the questionnaire and the other in the audio computer-assisted self-interviewing (ACASI) part. We used two different methods to identify likely underreporters of abortion and miscarriage: direct comparison of answers obtained from CAPI and ACASI and latent class models. The two methods produce very similar results. Although miscarriages are just as prone to underreporting as abortions, characteristics of women underreporting abortion differ somewhat from those misreporting miscarriages. Underreporters of abortions tended to be older, poorer, less likely to be Hispanic or Black, and more likely to have no religion. They also reported more traditional attitudes toward sexual behavior. By contrast, underreporters of miscarriage also tended to be older, poorer, and more likely to be Hispanic or Black, but were also more likely to have children in the household, had fewer pregnancies, and held less traditional attitudes toward marriage.

## Introduction

### Background

Survey respondents overreport socially desirable behaviors, like voting and donating money to charity, and underreport socially undesirable behaviors, like using illicit drugs or drinking too much (see [1] for a review). One behavior that generally is underreported in surveys is having had an abortion. In a series of studies, researchers have compared estimates of the number of abortions in the United States based on reports from the National Survey of Family Growth (NSFG) with estimates based on surveys of U.S. abortion providers. These studies have consistently shown that the NSFG estimates are too low—the survey respondents apparently report only about half of their abortions [2–6]. For example, a study [4] estimates that in the 2002 NSFG, respondents reported only 47 percent of their abortions. Similarly, Tourangeau and his colleagues used a sample that included women who were known to have had abortions and found that fewer than 75 percent of them reported *ever* having had an abortion and only about

**Data Availability Statement:** Data used in this manuscript are publicly available at study web site (https://www.cdc.gov/nchs/nsfg/index.htm).

**Funding:** The work reported here was supported by a grant (R01HD084473) from the Eunice

Kennedy Shriver National Institute of Child Health
& Human Development to The Guttmacher
Institute (Laura Lindberg, PI). The content is solely
the responsibility of the authors and does not
necessarily represent the official views of the
National Institutes of Health. The funders had no
role in study design, data collection and analysis,
decision to publish, or preparation of the
manuscript.

**Competing interests:** The authors have declared
that no competing interests exist.

half reported an abortion during the period in which they were known to have had one [7]. The problem of underreporting of abortions is not limited to the NSFG; abortions are also underreported in other national surveys in the U.S., including the 1997 National Longitudinal Survey of Youth and the Add Health Study, and in other countries [5, 8–11].

Less well studied is the underreporting about miscarriages. Even though both are pregnancy loss, miscarriage and abortion are differentially sensitive. Abortion is more stigmatized than miscarriage. One study [12] found about one in three respondents kept abortion as a secret whereas only 7% did so with miscarriage. In addition, of those who avoided telling about their abortion, 36% did so to avoid stigma. By contrast, less than 3% of respondents were concerned about stigma when they did not report their miscarriage. A second study compared [13] perception of abortion stigma to miscarriage stigma perception and found that the mean score on the abortion stigma perception is higher than that on the miscarriage stigma perception. Another study [14] further revealed that respondents erroneously perceived miscarriage as a rare complication of pregnancy. Those with a miscarriage felt that they lost a child and did something wrong, and felt guilty, alone, and ashamed [13, 14]. Abortions were sometimes misreported as miscarriage [8, 10]. There is limited evidence that miscarriage is also prone to misreporting [4, 15]. For instance, NSFG respondents were found to underreport miscarriage in the CAPI portion of the interview as often as they underreported abortion [15].

It is important to identify women respondents who are most prone to underreport abortions and miscarriages. A review of the literature on social desirability bias in survey reports argues that survey respondents often "edit" their answers prior to reporting them in order to avoid embarrassing themselves [1]. The review identifies several general things that affect whether survey reports will be subjected to such editing [1]. Respondents are more likely to misreport when an interviewer administers the questions than when the questions are self-administered; respondents in the socially undesirable category (e.g., non-voters) are much more likely to misreport than those in the desirable category (voters); and those who personally subscribe to the norms that make a given behavior socially desirable or undesirable are more likely to misreport than those who reject those norms. It is more embarrassing to admit something to an interviewer than to a computer and to have broken a norm than not to have broken one, especially when the norm is one that the respondent subscribes to.

We believe that some of the same factors that affect reports about other sensitive topics affect survey reports about abortion and miscarriage as well. For example, respondents are more likely to admit to having had an abortion when the questions are self-administered than when they are administered by an interviewer [8, 16, 17], although one exception is reported in [7]. Based on these findings, the NSFG asks two sets of questions about abortion, one set that is administered by the interviewers and a second set that is self-administered. The self-administered questions consistently elicit more reported abortions than the interviewer-administered ones [18].

This paper has three goals. The first is to evaluate the utility of the latent class analysis (LCA) approach in identifying underreporters of abortion and miscarriage. It is usually impossible to identify which respondents misreported from the survey data alone; ideally, medical records are needed for comparisons to survey data to properly identify underreporters. With NSFG, we can use answers from CAPI and ACASI to identify underreporters of abortion and miscarriage—that is, those who reported abortion in only one of the two modes. The weakness of this method is the failure to identify women who don't report abortion (or miscarriage) in either mode. LCA is a promising alternative because, in principle, it has the potential to identify those women who didn't report their abortion in either mode without needing medical records. We will replicate and extend the findings of an earlier analysis [18], using a larger and more recent data set and examining a wider range of LCA models on two pregnancy

outcomes. Then, to evaluate the utility of LCA models, we compare the results from LCA models to the results that use only survey data to identify likely underreporters.

The second goal is to try to understand factors affecting underreporting behavior of respondents by identifying the characteristics of respondents most prone to underreport abortion and miscarriage. Earlier work [4, 5] to characterize women underreporting abortion were conducted at the aggregate level and found that older women, married women, poorer women, college graduates, and Catholics are more likely to underreport abortions than their younger, unmarried, wealthier, less educated, and non-Catholic counterparts. We hope to replicate these findings and to extend them by using individual-level data and by examining additional characteristics of women associated with underreporting abortions. In particular, we test the idea that women from traditional cultural backgrounds, where disapproval of abortion is strong, are most prone to underreport their abortions.

The third goal is to investigate underreporting of miscarriages in the NSFG. We will compare the characteristics of women who underreport their abortions with those of women who underreport miscarriages to understand the differential sensitivity associated with abortion and miscarriage. We will examine whether or not there is a common set of characteristics that contribute to underreporting of both abortion and miscarriage.

## Methods

Our study is based on data from the National Survey of Family Growth (NSFG), a survey sponsored by the National Center for Health Statistics and, since 2002, carried out by the University of Michigan's Survey Research Center. The NSFG has been done periodically since 1973 and moved to a continuous design (with data collected every quarter) in 2006. We analyze data from the 2011 to 2015 NSFG; the samples during this period include data from 11,300 women respondents (and 9,321 men); our analysis is based solely on the women.

### Sample design

The target population for the 2011–2015 NSFG is the noninstitutionalized population 15–44 years old, whose usual place of residence is the 50 United States and the District of Columbia. It excludes people living in institutions, such as prisons or military bases.

To represent this population, the NSFG used a stratified five-stage area probability sample. The first stage of sample selection consisted of the selection of primary sampling units (PSUs); each PSU was a metropolitan area, a single county, or a group of counties. Prior to selection, all the PSUs were grouped into strata, based on census region and division, PSU size, and PSU Metropolitan Statistical Areas (MSA) / Non-MSA status. At the next stage, blocks or groups of adjoining blocks—second stage units (SSUs)—were selected. Both the PSUs and SSUs were selected with probability proportionate to size, where the size measure gave a higher selection probability to areas where at least 10 percent of the population was Black or Hispanic. In the third stage, a list of the housing units (HUs) within each SSUs was compiled and sample of HUs was selected. Interviewers either updated an existing list of addresses (based on the U.S. Postal Service's Delivery Sequence File) or created a list from scratch in SSUs where no list was available. The sample HUs were contacted and a short screening interview was administered. In units with eligible residents (that is, someone in the 15–44 age range), a fourth stage of selection was carried out that involved a random selection of one eligible person for the main interview. The within-household selection rates were set so that about 20 percent of all the interviews were with adolescent respondents (aged 15–19) and about 55 percent were with females. The final stage of sampling was carried out during the last two weeks of each quarter's

12-week field period; at this point, a subsample of the remaining cases was selected for continued follow-up.

## Data collection

Both the screener and most of the main NSFG interview were done via computer-assisted personal interviewing (CAPI); audio computer-assisted self-interviewing (ACASI) was used for the final section of the main interview, which contained the question items considered most sensitive (such as alcohol and drug use, involuntary sex, sexual disease, sexual orientation, and so on). The entire interview was programmed in the Blaise software (version 4.8). The ACASI portion of the interview featured text-to-speech—that is, a computer-generated voice—rather than a recorded human voice.

Under the NSFG's design, new samples are released each quarter. Interviewers attempt to complete all their assigned cases during the first ten weeks of each quarter; then, during the final two weeks of the 12-week field period, any remaining nonresponding cases are subsampled and interviewers attempt to complete the cases retained in the subsample. During this final phase of data collection each quarter, the incentive offered to respondents is doubled from $40 to $80.

## Key items

The questionnaire for female respondents consists of ten sections; the key abortion and miscarriage questions come in the second and tenth sections. In the second section, the respondent is asked whether she might be pregnant currently and how many times in total she has been pregnant. She is then asked about the outcome of each pregnancy (miscarriage, stillbirth, abortion, ectopic or tubal pregnancy, or live birth) and the month and year when the pregnancy ended. From these items, derived variables are constructed, indicating the number of abortions and miscarriages the respondent reported in the past five years under CAPI. At the beginning of the final section of the questionnaire, respondents are asked in separate questions (this time in ACASI), how many pregnancies they had in the last five years that ended in a live birth; a stillbirth, miscarriage, or tubal or ectopic pregnancy; or an abortion. The data from the second and the ACASI sections of the questionnaire are recoded into dummy variables denoting a report of at least one abortion (or miscarriage) in the last five years. They are then used as the main indicators for the LCA models. Table 1 shows the (unweighted) cross-tabulation of the key indicators of abortion and miscarriage reporting.

## Analytical strategy

We first used only survey reports to identify consistent reporters of abortion as those who reported abortion in both modes and likely underreporters of abortion as those who reported

**Table 1. Unweighted cross-tabulation of the ACASI and CAPI indicators of abortion and miscarriage.**

|  |  | CAPI Indicator | |
|---|---|---|---|
|  |  | YES (Abortion Reported) | NO (No Abortion Reported) |
| **ACASI Indicator** | YES (Abortion Reported) | 422 | 273 |
|  | NO (No Abortion Reported) | 39 | 10544 |
|  |  | CAPI Indicator | |
|  |  | YES (Miscarriage Reported) | NO (No Miscarriage Reported) |
| **ACASI Indicator** | YES (Miscarriage Reported) | 840 | 506 |
|  | NO (No Miscarriage Reported) | 69 | 9863 |

abortion in only one mode. As shown in Table 1, 422 women are classified as consistent reporters of abortion and 312 (= 39+273) women as likely underreporters of abortion. In the same manner, 840 women are classified as consistent reporters of miscarriage and 575 (= 69 +506) women as likely underreporters of miscarriage. For the regression analyses, we assume that consistent reporters of abortion and miscarriage are truthful reporters of abortion and miscarriage.

Then, we adopted a three-step approach to apply LCA in the analysis. LCA models the relationships among a set of observed categorical variables (in this case, the CAPI and ACASI "indicators" of abortion/miscarriage) measuring one unobserved (that is, "latent") categorical variable with two or more classes (in this case, a two-class latent abortion/miscarriage variable). The associations between the observed variables reflect the fact that the population consists of a set of mutually exclusive and exhaustive latent classes with different distributions on the observed variables. Within each of the latent classes, the observed variables are unrelated. It is this key assumption of "local independence" that allows inferences about the latent class variable [19].

In such a model, the probability of an observed response ($\mu_j$) on question $j$ depends on the conditional probability of observing that response given that the respondent is in latent class $k$, summed across all $K$ of the latent classes. Given that the responses are independent of each other within each latent class, the probability of the vector of responses $\mathbf{\mu}$ is:

$$\mathbf{\mu} = \sum_{k=1}^{K} P(c = k)\prod_{j=1}^{J} P(\mu_j|c = k),$$ (1)

in which there are $K$ latent classes, each with a "prevalence" (i.e., unconditional probability) of $P(c = k)$. The model produces estimates of these unconditional probabilities—representing the relative sizes of each latent class—as well as of the conditional probabilities of each response within each latent class ($P(\mu_j|c = k)$).

For a two-class LCA, three indicators are necessary for the model to be identified [19]. When there are only two available indicators, researchers can choose to impose various restrictions on the LCA model parameters to achieve identifiability. For example, the false positive probability might be assumed to be zero or the latent classes might be assumed to be the same size. (Additional examples are provided in [20]). However, such assumptions are often implausible. Another approach is to include a grouping variable ($G_i = 1, 2,\ldots, g$) in the model that predicts membership in the latent class, as in the Hui-Walter model [21–24]. To achieve an identifiable model, the Hui-Walter model makes two assumptions about the grouping variable:

1. The prevalence rates differ by the level of the grouping variable (the unequal prevalence assumption); and

2. The false positive and false negative probabilities are the same in each level of the grouping variable (the equal error probabilities assumption). (False positive error arises when a respondent is assigned to the 'did not have an abortion' latent class but she reported having had one, whereas a false negative error occurs when a respondent is assigned to the 'had an abortion' latent class but did not report having one.)

In this paper, we first fit a two-class latent class model. Since there were only two indicators of abortion/miscarriage, we adopted the Hui-Walter approach and used marital status as a grouping variable to achieve an identifiable model. The model also included additional covariates, such as age (20 to 29 years old versus all others), poverty level (three classes); and race/ethnicity (Black or Hispanic women versus all others). In addition, we explored models that

included a third indicator of abortion/miscarriage–whether the respondent reported in the CAPI section of the questionnaire ever being pregnant. We used PROC LCA in SAS to fit the models [25].

Next, we assigned each respondent to one of the two latent classes (women who had had an abortion in the last five years and those who had not) based on their posterior class membership probabilities. We used two different methods to make these assignments. In the first method, modal assignment, each respondent was assigned to the latent class to which she had the highest probability of belonging. In the second method, stochastic assignment, the respondent was randomly assigned to a latent class, with the assignment probability equal to her posterior class membership probability.

We then compared respondents' reports of their abortion status to their predicted class membership. We divided the respondents into four groups: 1) those who were assigned to the class of women who had had an abortion in the last five years and reported it in both the CAPI and ACASI portions of the questionnaire; 2) those who were assigned to that class but did *not* report an abortion in at least one mode; 3) those who are assigned to class of women who had not had an abortion in the last five years, but reported an abortion in at least one mode; and 4) those who were assigned to the class who had not had an abortion in the last five years and did not report an abortion in either mode. The first group of respondents are truthful reporters of abortion (according to the model) whereas the second group are underreporters of an abortion because they failed to report having an abortion in either CAPI or ACASI. The third group, according to the model, represents those who overreport abortion and the fourth group, accurate reporters of no abortion.

In the third step, we fit logistic regression models to compare those who underreported their abortions in at least one mode (i.e., reporting group 2 above) to truthful reporters of abortions (i.e., reporting group 1) on a wide range of variables related to abortion. Predictors include age (as a continuous variable), marital status (a dummy variable contrasting "married or cohabitating" vs. all others), race and ethnicity (a dummy variable contrasting "Hispanic or Non-Hispanic Black women" vs. all others), total family income recoded into 15 income brackets, whether there were no children under the age of 18 in the household (= 1) or at least one (= 0), the number of pregnancies in the lifetime (as a continuous variable), the number of lifetime male sexual partners (as a continuous variable), whether the respondent currently reported no religion (= 1) or any religion (= 0), whether the respondent lived in a metropolitan area (= 1) or not (= 0), whether the respondent' mother had a high school or less education (= 1) or more than high school (= 0), whether the respondent was born outside of USA (= 1) or not (= 0), and whether the respondent completed the ACASI interview in Spanish (= 1) or not (= 0).

We also created three scales for possible inclusion in the models. The first ("Risky substance use behaviors") was a count of the number of risky substance use behaviors the respondent reported, including having smoked at least 100 cigarettes in her lifetime; having drunk beer and other alcoholic beverages; smoked marijuana; used cocaine, crack, Crystal or meth; and injected drugs other than prescriptions at least once or twice during the year. The second and third scales were based on a battery of attitudinal items on sex, divorce, and homosexuality. All of these items used a five-point agree-disagree response scale. We created a scale of traditional sexual attitudes, based on the respondent's answers to eight of the items (e.g., *Sexual relations between two adults of the same sex is all right*). These eight items loaded highly (absolute value of .55 or higher) on the first component of an exploratory factor analysis of the items. [S1 Table] gives the exact wording of all eight items and their loadings on this scale. Higher scores on the index indicate more traditional attitudes. Our hypothesis was that the women who had more traditional attitudes would be more reluctant to report an abortion and thus more likely

to be underreporters. The factor analysis also yielded a second factor, which we labelled attitudes toward marriage. This scale was based on answers to three of the items (e.g., *Divorce is usually the best solution when a couple can't seem to work out their marriage problems*). Again, higher scores indicate more traditional attitudes. [S2 Table] gives the exact wordings for all three of these items and their loadings on this scale. We thought that woman with more negative attitudes toward marriage might be more likely to report their abortions.

The logistic regression models were run in SAS (PROC SURVEYLOGISTIC), accounting for the complex sample design (that is, the weights, stratification, and clustering).

We used the same analytic strategy to identify and predict underreporters of miscarriage.

## Results

### Predicted latent classes for abortion and miscarriage reporting

Table 2 displays the prevalence of the two latent abortion classes produced by the two-indicator LCA that drew only on the CAPI and ACASI responses and by the three-indicator LCA that included whether respondents reported they were ever pregnant (= 1) or not (= 0) as the third indicator. As noted earlier, we used both the deterministic (assignment to the more likely latent class) and stochastic approaches (assignment based on the predicted posterior probabilities). The results are quite consistent across the LCA models and class assignment approaches. From 6.2% (n = 703) to 6.6% (n = 741) of the women were assigned to the "had an abortion" class, which is a bit higher than the proportion of self-reported abortions in either CAPI or ACASI modes, as shown in Table 1. The weighted figures are similar. Not surprisingly, LCA models show that the CAPI indicator of abortion has, on average, a higher false negative rate than the ACASI indicator (37% vs. 8%); in other words, more women who were assigned to the "had an abortion" latent class reported *not* having an abortion in the CAPI mode than in the ACASI mode. This is consistent with past findings [18] that CAPI produces more underreports of having had an abortion than ACASI.

Results on the prevalence estimates of the two latent miscarriage classes are similar across LCA models and class assignment methods. From 12.2% (n = 1,379) to 12.6% (n = 1,415) of women respondents were assigned to the "had a miscarriage" latent class. This is also higher than the unweighted percent of women reporting having had a miscarriage in either mode of data collection. Similarly, the CAPI indicator of miscarriage also has a higher false negative rate (34%) than the ACASI indicator (7%).

Overall, the findings were similar whether the latent class model was based on the Hui-Walter assumptions or incorporated a third indicator, and whether the class membership was assigned in a deterministic or stochastic manner.

**Table 2. Predicted latent abortion and miscarriage classes (unweighted).**

| | 2-indicator LCA | | 3-indicator LCA | |
|---|---|---|---|---|
| **Latent Abortion Class** | Modal Assignment | Random Assignment | Modal Assignment | Random Assignment |
| Had an abortion in last five years | 734 (6.5%) | 741 (6.6%) | 715 (6.3%) | 703 (6.2%) |
| Did not have an abortion in last five years | 10,544 (93.5) | 10,537 (93.4) | 10,563 (93.4) | 10,575 (93.8) |
| Total | 11,278 (100%) | 11,278 (100%) | 11,278 (100%) | 11,278 (100%) |
| | **2-indicator LCA** | | **3-indicator LCA** | |
| **Latent Miscarriage Class** | Modal Assignment | Random Assignment | Modal Assignment | Random Assignment |
| Had a miscarriage in last five years | 1,415 (12.6%) | 1,379 (12.2%) | 1,393 (12.3%) | 1,395 (12.4%) |
| Did not have a miscarriage in last five years | 9,863 (87.4) | 9,899 (87.8) | 9,885 (87.7) | 9,883 (87.7) |
| Total | 11,278 (100%) | 11,278 (100%) | 11,278 (100%) | 11,278 (100%) |

**Table 3. Abortion reporting behaviors (unweighted).**

| | 2-indicator LCA | | 3-indicator LCA | |
|---|---|---|---|---|
| | Modal Assignment | Stochastic Assignment | Modal Assignment | Stochastic Assignment |
| Truthful reporters of abortion | 422 | 422 | 422 | 422 |
| Underreporters of abortion | 312 | 319 | 293 | 281 |
| Underreporters as percent of those classified as having had an abortion | 42.5% | 43.0% | 41.0% | 40.0% |
| Overreporters of abortion | 0 | 18 | 19 | 52 |
| Truthful reporters of no abortion | 10,544 | 10,519 | 10,544 | 10,523 |

**Note**: Figures are unweighted.

### Underreporters versus truthful reporters of abortion

We compared respondents' self-reports of abortion to the latent class assigned, placing respondents in one of the four reporting groups. As shown in Table 3, the results are again quite consistent across two- and three-indicator LCA models and across the two methods of assigning respondents to a latent class. Regardless of the LCA model or assignment approach, 422 respondents are classified as truthful reporters of abortion; they reported they had at least one abortion in the last five years in both CAPI and ACASI. Close to 40% of respondents in the latent class of women who had an abortion (n = 281 to 319) are classified as underreporters of abortion; they denied having had an abortion in the last five years in least one of the two modes. The weighted figures are similar; from 43 to 47 percent of women underreport their abortions in at least one mode. Recall Table 1, 422 women are classified as truthful reporters of abortion and 312 women as underreporters of abortion using only survey data. The different methods of identifying underreporters of abortion produce very similar results.

Only a handful of respondents are classified as overreporters of abortion. The vast majority of respondents who reported no abortion in either mode were classified as truthful reporters of no abortion.

We ran logistic regression models predicting the respondent's likelihood of being classified as an underreporter (group 2 in Table 3) rather than as a truthful reporter of an abortion in the past five years (group 1 in Table 3). Table 4 presents model results in logit scale whereas [S3 Table] provides odds ratios, 95% confidence interval, and the *p*-values. As shown in Table 4, three variables were significantly related to a woman's likelihood to underreport her abortion across all five of the models. Older women were more likely to underreport an abortion. By contrast, women with no religion and with higher incomes were *less* likely to underreport an abortion. Traditional sexual attitudes were positively associated with the women's likelihood to underreport an abortion; the association was statistically significant in four of the five models and was marginally significant in the remaining model. The number of risky substance use behaviors the respondent reported was negatively related to the women's likelihood to underreport an abortion. However, this relationship was statistically significant at $p < .05$ for three models and marginally significant under two more ($p < .10$). Finally, Hispanic or Black women were significantly less likely to underreport an abortion in two of the five models. It is worth mentioning that the model results using LCA classification are highly consistent with the model results using the classification based solely on the survey data.

### Underreporters versus truthful reporters of miscarriages

Although miscarriages are presumably less sensitive to report than abortions, they are not always reported accurately. Table 5 shows the proportion of women classified in each

**Table 4. Logistic regression coefficients in models predicting underreporting of abortion.**

| Parameter | 2-indicator LCA, modal assignment | | 2-indicator LCA, random assignment | | 3-indicator LCA, modal assignment | | 3-indicator LCA, random assignment | | Survey Data Only | |
|---|---|---|---|---|---|---|---|---|---|---|
| | Estimate | SE | Estimate | SE | Estimate | SE | Estimate | SE | Estimate | SE |
| Intercept | 2.06 | 0.57 | 2.39 | 0.57 | 1.76 | 0.55 | 1.87 | 0.55 | 2.06 | 0.57 |
| Age (centered at mean) | **0.07**** | 0.02 | **0.07*** | 0.02 | **0.09**** | 0.02 | **0.11***** | 0.02 | **0.07**** | 0.02 |
| Married or Cohabitating | 0.15 | 0.26 | 0.17 | 0.24 | 0.37 | 0.27 | 0.49 | 0.26 | 0.15 | 0.26 |
| Hispanic or Black | **-0.55*** | 0.27 | -0.49 | 0.28 | -0.37 | 0.29 | -0.45 | 0.31 | **-0.55*** | 0.27 |
| No Children in Household | -0.27 | 0.27 | -0.31 | 0.27 | -0.51 | 0.30 | **-0.62** | 0.29 | -0.27 | 0.27 |
| Number of Pregnancies | -0.10 | 0.07 | -0.15 | 0.08 | -0.08 | 0.07 | -0.11 | 0.07 | -0.10 | 0.07 |
| Number of Life Partners | -0.01 | 0.01 | -0.01 | 0.01 | 0.01 | 0.01 | 0.01 | 0.01 | -0.01 | 0.01 |
| No Religion | **-0.78**** | 0.24 | **-0.92**** | 0.23 | **-0.56*** | 0.24 | **-0.49*** | 0.24 | **-0.78**** | 0.24 |
| Total Income | **-0.09**** | 0.03 | **-0.08*** | 0.03 | **-0.10**** | 0.03 | **-0.10**** | 0.03 | **-0.09**** | 0.03 |
| Metropolitan Area | -0.51 | 0.28 | -0.51 | 0.28 | -0.54 | 0.28 | -0.57 | 0.29 | -0.51 | 0.28 |
| Mother with High School Education or Less | 0.01 | 0.25 | 0.06 | 0.25 | 0.03 | 0.29 | 0.09 | 0.29 | 0.01 | 0.25 |
| Born outside USA | 0.08 | 0.41 | -0.04 | 0.43 | -0.06 | 0.34 | -0.21 | 0.38 | 0.08 | 0.41 |
| Interview Language | 0.48 | 0.70 | 0.43 | 0.70 | 0.30 | 0.69 | 0.23 | 0.73 | 0.48 | 0.70 |
| Risky Substance Use Behaviors | -0.25 | 0.15 | **-0.35*** | 0.17 | **-0.30*** | 0.14 | **-0.37*** | 0.15 | -0.25 | 0.15 |
| Traditional Sexual Attitudes | **0.39*** | 0.17 | 0.29 | 0.17 | **0.45*** | 0.18 | **0.41*** | 0.18 | **0.39*** | 0.17 |
| Attitudes toward Marriage | 0.04 | 0.16 | 0.01 | 0.17 | -0.02 | 0.18 | -0.07 | 0.19 | 0.04 | 0.16 |
| n | 696 | | 702 | | 678 | | 665 | | 696 | |
| Pseduo-R | 0.1846 | | 0.1867 | | 0.2146 | | 0.2301 | | 0.1846 | |

Note:

* $p<0.05$

** $p<0.01$

*** $p<0.001$

reporting group for miscarriage. Regardless of the LCA model or assignment approach, 840 respondents are classified as truthful reporters of miscarriage; they reported having at least one miscarriage in the last five years in both CAPI and ACASI. Close to 40% of respondents in the latent class of women who had a miscarriage (n = 539 to 575) are classified as underreporters of miscarriage; they denied having had a miscarriage in the last five years in least one of the two modes. The weighted figures are similar; from 35 to 36 percent of women underreport their miscarriages in at least one mode. Table 5 shows that miscarriages are almost as prone to underreporting as abortions, a finding consistent with the previous findings [15]. Overreporters of miscarriages are, according to the latent class models, very rare. This classification pattern is very similar to the classifications based on the survey data only (see Table 1).

To what extent is underreporting of miscarriages associated with the same variables as underreporting of abortion? Table 6 presents model results in logit scale whereas [S4 Table]

**Table 5. Miscarriages reporting behaviors (unweighted).**

| | 2-indicator LCA | | 3-indicator LCA | |
|---|---|---|---|---|
| | Modal Assignment | Stochastic Assignment | Modal Assignment | Stochastic Assignment |
| Truthful reporters of miscarriage | 840 | 840 | 840 | 840 |
| Underreporters of miscarriage | 575 | 539 | 553 | 555 |
| Underreporters as percent of those classified as having had a miscarriage | 40.6% | 39.1% | 39.7% | 39.8% |
| Overreporters of miscarriage | 0 | 78 | 22 | 57 |
| Truthful reporters of no miscarriage | 9,863 | 9,821 | 9,863 | 9,826 |

Table 6. Logistic regression coefficients in models predicting underreporting of miscarriage.

| Parameter | 2-indicator LCA, Modal Assignment | | 2-Indicator LCA, Random Assignment | | 3-indicator LCA, Modal Assignment | | 3-indicator LCA, Random Assignment | | Survey Data Only | |
|---|---|---|---|---|---|---|---|---|---|---|
| | Estimate | SE | Estimate | SE | Estimate | SE | Estimate | SE | Estimate | SE |
| Intercept | 0.64 | 0.38 | 0.69 | 0.40 | 0.41 | 0.38 | 0.52 | 0.39 | 0.64 | 0.38 |
| Age (centered at mean) | **0.06**\*\*\* | 0.01 | **0.05**\*\* | 0.02 | **0.07**\*\*\* | 0.02 | **0.08**\*\*\* | 0.02 | **0.06**\*\* | 0.01 |
| Married or Cohabitating | **-0.46**\* | 0.20 | -0.29 | 0.21 | -0.26 | 0.20 | -0.18 | 0.21 | **-0.46**\* | 0.20 |
| Hispanic or Black | **0.43**\* | 0.18 | **0.53**\* | 0.19 | **0.52**\*\* | 0.18 | **0.54**\*\* | 0.18 | **0.43**\* | 0.18 |
| No Children in Household | **-0.69**\*\* | 0.23 | **-0.61**\* | 0.22 | **-0.91**\*\*\* | 0.25 | **-1.04**\*\*\* | 0.26 | **-0.69**\*\* | 0.23 |
| Number of Pregnancies | **-0.21**\*\* | 0.06 | **-0.26**\*\*\* | 0.07 | **-0.17**\*\* | 0.06 | **-0.23**\*\* | 0.07 | **-0.21**\*\* | 0.06 |
| Number of Life Partners | -0.02 | 0.01 | -0.02 | 0.01 | -0.01 | 0.01 | -0.01 | 0.01 | -0.02 | 0.01 |
| No Religion | -0.06 | 0.22 | -0.05 | 0.22 | -0.02 | 0.22 | 0.02 | 0.22 | -0.06 | 0.22 |
| Total Income | **-0.06**\*\* | 0.02 | **-0.06**\*\* | 0.02 | **-0.07**\*\* | 0.02 | **-0.08**\*\* | 0.02 | **-0.06**\*\* | 0.02 |
| Metropolitan Area | 0.29 | 0.18 | 0.23 | 0.19 | 0.26 | 0.18 | 0.27 | 0.19 | 0.29 | 0.18 |
| Mother with High School Education or Less | 0.05 | 0.18 | 0.14 | 0.18 | 0.02 | 0.18 | 0.13 | 0.18 | 0.05 | 0.18 |
| Born outside USA | 0.09 | 0.26 | 0.06 | 0.26 | 0.10 | 0.27 | 0.08 | 0.26 | 0.09 | 0.26 |
| Interview Language | 0.10 | 0.42 | 0.00 | 0.43 | -0.04 | 0.43 | -0.10 | 0.45 | 0.10 | 0.42 |
| Risky Substance Use Behaviors | 0.13 | 0.11 | 0.06 | 0.11 | 0.10 | 0.11 | 0.11 | 0.10 | 0.13 | 0.11 |
| Traditional Sexual Attitudes | 0.01 | 0.12 | -0.05 | 0.12 | -0.01 | 0.13 | -0.06 | 0.13 | 0.01 | 0.12 |
| Attitudes toward Marriage | **-0.34**\* | 0.13 | **-0.36**\* | 0.13 | **-0.37**\* | 0.13 | **-0.39**\* | 0.13 | **-0.34**\* | 0.13 |
| n | 1351 | | 1316 | | 1330 | | 1330 | | 1351 | |
| Pseduo-R | 0.0992 | | 0.0973 | | 0.1051 | | 0.1164 | | 0.0992 | |

Note:

\* $p < 0.05$

\*\* $p < 0.01$

\*\*\* $p < 0.001$

provides odds ratios, 95% confidence interval, and the *p*-values. As shown in Table 6, some variables, such as age and income are significantly related to underreporting both abortions and miscarriages. The odds of underreporting increases with age, but is lower for women with higher incomes. Still, there are some noteworthy differences in the predictors of underreporting of the two outcomes. Hispanic or Black women are significantly more likely to underreport miscarriages than other women, but not abortions (where the sign for this variable is in the opposite direction). Women with no children and women with more pregnancies are significantly associated with less underreporting of miscarriage, but the number of children and the number of pregnancies had no association with underreporting of abortion. Furthermore, women holding more traditional attitudes toward marriage were less likely to underreport miscarriages. Perhaps the key differences across the models are that risky substance use behaviors and the traditional sexual attitudes are significantly related only to abortion underreporting.

Models results from LCA classifications are, again, very similar to results from classifications using survey data, with only two exceptions. The effects of marital status on miscarriage underreporting are statistically significant for only two of the four LCA models.

## Discussion

Because of the social stigma associated with abortion, women are found to underreport abortions in surveys [e.g., 4, 5]. Although the use of ACASI reduces the extent of underreporting, it does not eliminate it. It is, therefore, important to know which respondents are more likely to

underreport their abortions. We used latent class analysis (LCA) to compare underreporters of abortions to truthful reporters. The key advantage of LCA lies in the fact that it does not require an error-free indicator of abortion. In this paper, we used LCA to predict the probabilities that respondents from the NSFG fell into one of two latent classes—woman who had had an abortion in the last five years and those who had not. We then used these probabilities to assign women to a latent class. We ran LCA models that used two or three indicators and we used two different methods to assign respondents to a latent class. The results are quite consistent across the different models and methods of assignment. About 7 percent of the women are assigned to the latent class of women who had had an abortion in the last five years and, among them, about 40 percent did not report having had an abortion in at least one mode. This level of underreporting is consistent with other research, though lower than the rates of underreporting found in comparisons to the Guttmacher Institute abortion providers census [4, 5].

Among those classified as having had an abortion, we distinguished underreporters from truthful reporters of abortion, based on their survey answers. We examined their demographic characteristics, fertility characteristics, and other characteristics (whether the respondent lived in a metropolitan area, whether the respondent's mother had high school or less education, whether the respondent had a religion, and whether or not the respondent completed the ACASI interview in Spanish). In addition, we constructed three indices to measure the respondent's engagement in risky behaviors, traditional attitudes toward sex, and attitudes toward marriage. Our modeling efforts were guided by past results and by the general hypothesis that women from traditional cultural backgrounds, featuring unfavorable attitudes toward abortion, would be more likely to underreport their abortions.

The models consistently show that older women, women with lower household income, women with a religion, and women with traditional attitudes were more likely to underreport abortions. These findings held whether we classified women based on the latent class models or simply based on their inconsistent responses across modes. In general, these findings are consistent with studies comparing reports from the National Survey of Family Growth with external "gold standard" counts. For example, one study [5] found that women who were low income, Catholic, born in the U.S., and reported their race as "Other" reported a lower proportion of their abortions.

We also fit similar models for reports about miscarriages. Although miscarriage was found as prone to underreporting as abortion, by contrast with abortion, underreporting of miscarriages was *not* associated with having a religion, risky substance use behaviors, or traditional sexual attitudes. It seems likely that, although miscarriages may be painful to recall and or embarrassing for many women, they are not stigmatized by traditional attitudes on sexual behavior. We also found that women with at least one child under the age of 18 in their household, women with fewer pregnancies, and women with less traditional attitudes towards marriage were more likely to underreport miscarriages as compared to other respondents, but no more likely to underreport their abortions. Race/ethnicity was the one variable related in the opposite ways to reporting on the two pregnancy outcomes. Black and Hispanic women were more likely to underreport a miscarriage than other women, but less likely to underreport an abortion (though the latter effect is not significant in every model). Perhaps miscarriage carries greater stigma in these minority communities.

The results suggest that underreporting of abortions is driven by the social stigma associated with abortion rather than, say, recall error. We did not find that women with *fewer* pregnancies were less likely to underreport abortions; since recall is likely to be easier for these respondents, this finding suggests that underreporting is not due to forgetting or difficulty dating an abortion event. However, an earlier study [13] found evidence suggesting that both recall and the social stigma play a role in underreporting of abortions.

We included two variables in our model in an attempt to clarify how attitudinal factors influence the dynamics of underreporting. One variable was based on the number of risky substance use behaviors (such as smoking 100 cigarettes, using cocaine, or injected drugs) each woman reported; the other was based on a series of attitudinal items assessing attitudes about sex, divorce, and gays. We thought that the two variables would be an indirect indicator of the extent to which women subscribed to the norm that abortion is wrong (which is not measured directly in the NSFG). Both variables were related to abortion reports but not with miscarriage reports, suggesting that how a woman feels about abortion is an important factor in her willingness to report one. However, it is also possible that the risky substance use behavior scale is an indicator of a willingness to report sensitive behaviors in general so that willingness to report illicit substance use may be correlated with willingness to report abortion. To accurately capture abortion, survey researchers may have to find a way to make the stigma associated with abortion less salient, thereby making it easier for people to admit having had one. Possible questionnaire design strategies include the use of random response technique [9], ballot-box technique [26], forgiving wordings [1, 27], and so on.

To evaluate the utility of using LCA to identify underreporters of abortion and miscarriage, we drew on survey reports directly to classify people into truthful reporters of abortion and underreporters of abortion. This type of classification has obvious problems since women could deny having an abortion in both CAPI and ACASI. We suspect that women denying having an abortion in both modes find questions about abortions more sensitive and stigmatic than women who reported having an abortion in the ACASI mode but not in the CAPI mode. Still, the classification results and modeling results are highly consistent with the results based on LCA model. However, LCA doesn't seem to offer much of an advantage in exploring underreporting over the simpler method of using survey data only to identify inconsistent reporters. We suspect that, our models, like all LCAs, rely on assumptions to make estimation mathematically possible. For instance, the model in Eq 1 assumes local independence; that is, each of the indicator variables is fallible, but the errors associated with each one are uncorrelated within the latent class. Although this assumption is sometimes relaxed [20], most applications of LCA to assess errors in survey variables are based on the assumption of conditional independence [20–22]. However, in practice, respondents' answers to one question about a topic may influence their responses to a second question measuring the same construct. They may, for example, recall their earlier answers and try to avoid appearing inconsistent. When assumptions are not met in practical survey situations, LCA estimates of error rates and prevalence are questionable [18, 28]. In the current case, the models seem to underestimate the prevalence of underreporters.

We do not see any magic bullet for markedly improving reporting about abortion or miscarriage, but do advocate including relevant attitudinal variables in future studies that attempt to measure these pregnancy outcomes. The women who underreport these outcomes are women who are likely to find them embarrassing or shameful.

Even though LCA does not seem to offer much leverage in identifying underreporters of abortion and miscarriage in the current case, it is still a useful tool and has been used for other surveys such as the Current Population Survey [21] and for other topics such as drug use [22, 23]. We encourage researchers to continue exploring and evaluating the use of LCA when there are no gold standards or error-free records available.

## Supporting information

**S1 Table. Question wording and factor loading for items in the traditional sex attitudes scale.**
(DOCX)

**S2 Table. Question wording and factor loading for items in the attitudes toward marriage scale.**
(DOCX)

**S3 Table. Results of logistic regression models predicting underreporting of abortion.**
(DOCX)

**S4 Table. Results of logistic regression models predicting underreporting of miscarriage.**
(DOCX)

## Acknowledgments

We are grateful to Laura Lindberg and Isaac Maddow-Zimet for providing recodes of the key variables and for their comments on an earlier draft of this paper.

## Author Contributions

**Conceptualization:** Ting Yan, Roger Tourangeau.

**Data curation:** Ting Yan.

**Formal analysis:** Ting Yan.

**Methodology:** Ting Yan, Roger Tourangeau.

**Resources:** Ting Yan.

**Software:** Ting Yan.

**Validation:** Ting Yan.

**Visualization:** Ting Yan.

**Writing – original draft:** Ting Yan, Roger Tourangeau.

**Writing – review & editing:** Ting Yan, Roger Tourangeau.

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
