## [Decision Letter · Decision Letter 0]

17 Feb 2022

PONE-D-21-29890Underreporting abortions and miscarriagesPLOS ONE

Dear Dr. Yan,

Thank you for submitting your manuscript to PLOS ONE. After careful consideration, we feel that it has merit but does not fully meet PLOS ONE’s publication criteria as it currently stands. Therefore, we invite you to submit a revised version of the manuscript that addresses the points raised during the review process.

We look forward to receiving your revised manuscript.

Kind regards,

Janet E Rosenbaum, Ph.D.

Academic Editor

PLOS ONE

Journal Requirements:

The work reported here was supported by a grant (R01HD084473) from the Eunice Kennedy Shriver National Institute of Child Health & Human Development to The Guttmacher Institute (Laura Lindberg, PI). The content is solely the responsibility of the authors and does not necessarily represent the official views of the National Institutes of Health. 

The work reported here was supported by a grant (R01HD084473) from the Eunice Kennedy Shriver National Institute of Child Health & Human Development to The Guttmacher Institute (Laura Lindberg, PI). The content is solely the responsibility of the authors and does not necessarily represent the official views of the National Institutes of Health. 

The work reported here was supported by a grant (R01HD084473) from the Eunice Kennedy Shriver National Institute of Child Health & Human Development to The Guttmacher Institute (Laura Lindberg, PI). The content is solely the responsibility of the authors and does not necessarily represent the official views of the National Institutes of Health. 

No authors have competing interests

Reviewers' comments:

Reviewer's Responses to Questions

**Comments to the Author**

1. Is the manuscript technically sound, and do the data support the conclusions?

Reviewer #1: Partly

Reviewer #2: Yes

Reviewer #3: No

Reviewer #4: Yes

2. Has the statistical analysis been performed appropriately and rigorously? 

Reviewer #1: No

Reviewer #2: Yes

Reviewer #3: No

Reviewer #4: Yes

3. Have the authors made all data underlying the findings in their manuscript fully available?

Reviewer #1: Yes

Reviewer #2: Yes

Reviewer #3: Yes

Reviewer #4: Yes

4. Is the manuscript presented in an intelligible fashion and written in standard English?

Reviewer #1: Yes

Reviewer #2: Yes

Reviewer #3: Yes

Reviewer #4: Yes

5. Review Comments to the Author

Reviewer #1: The paper seeks to contribute to our knowledge of the patterns and overall trends in underreporting of abortions and miscarriages in survey research. The authors use the National Survey of Family Growth (NSFG) to undertake their analyses. They leverage the multiple modes of data collection to investigate the characteristics of women who appear to underreport miscarriages and abortions. In addition, the authors propose using latent class analysis as a method for predicting women who are likely underreporting abortions and miscarriages. The authors use these results to compare the characteristics of women who appear to underreport miscarriages and abortions. Use of LCA and the assessment of miscarriage underreporting in the NSFG are notable contributions of the paper.

This paper has two key strengths. First, the issue of miscarriage underreporting is highly relevant and tends to be understudied due to the lack of external estimates for the “true” number of miscarriages in a population. The paper’s approach to this problem using the two forms reporting is novel and provides important insights into a topic that is difficult to study and quantify. Second, the paper’s use of LCA to identify possible under-reporters is novel and a potentially useful methodological advance for the field, especially for miscarriage, where external data are limited.

Although the topics under study are of importance and the paper makes advances in the methodological issues with measurement of these topics, the paper has two central weaknesses that limit it from fully or convincingly reaching its high potential. First, the paper is confusingly framed and situated, especially as it relates to the inclusion of the analyses related to abortion. For example, a great deal is known about the characteristics of women who underreport abortions, including specifically in the NSFG as the paper cites in the introduction. Yet, the paper motivates its analysis by suggesting little is known about the characteristics of these women. While the paper uses individual-level data, which allows for the analysis of new or additional characteristics associated with underreporting, this nuance is not clear in their introduction. More generally, the introduction and background sections of the paper are disconnected and difficult to follow (specific comments related to this can be found in the minor or stylistic issues section following these comments). Relatedly, it is unclear why the paper addresses both abortion and miscarriage underreporting. As aforementioned, abortion underreporting is well-documented, especially in the NSFG. The analysis of miscarriages, however, is quite novel. Thus, the framing of the paper is confusing and does not seem to adequately review the literature related to abortion underreporting nor make a convincing argument or present a clear rationale for why these issues are discussed in-tandem in the paper.

Second, the paper oversells the meaning of the differences between the CAPI and ACASI comparisons. That is, the paper suggests that inconsistent reporting is equivalent to underreporting, yet, there may well be something substantively different between women who never report an abortion they have had and those who sometimes will report it (on CAPI but not ACASI). There seems to be no way to know if these women are qualitatively similar or if we can draw conclusions about under-reporters as an entire group based on these inconsistent reporters. This approach is novel and interesting, but it seems to fall especially short in the case of abortion where external estimates to quantify underreporting exist. In other words, if the paper were solely focused on miscarriage, it would be more methodologically justifiable to use this approach (since a better alternative is currently lacking). Additionally, the comparison of the LCA to the inconsistent reporters is also less compelling considering these issues.

In addition to the more major issues outlined above, I also provide feedback on minor or stylistic issues by section below.

Introduction

- It is not clear why the section on LCA is presented in the introduction/background section. It seems that the materials here would be more appropriate in the analytic method section, and the limitations of LCA could be explored in the discussion section.

- The structure of the introduction could be revised for clarity as well as to more specifically discuss underreporting of abortion and miscarriages.

Methods

- It would be more accurate to label “truthful” reporter “consistent” reporters While it is unlikely someone would report an abortion or miscarriage they didn’t have, consistent reporters more accurately describes what is being observed (consistency between CAPI and ACASI)

- The choice to collapse Black and Hispanic populations together should be explicated, especially considering the import of language and religiosity in the findings. It would be stronger to separate these groups, if there is sufficient power

Results

- More descriptive table names would be helpful, especially for Table 5

- Person-first or person-centered language should be used (i.e., Black women or Black respondents or respondents of color, etc. rather than “Blacks”)

Discussion

- This section would be stronger with less reiteration of the findings and more exploration of the patterns observed and contribution

- It seems out of place to have a reference to a table in the discussion

- Inclusion of citations and references to work focused on how to improve abortion reporting or reporting of sensitive questions would be useful in the discussion

Overall, the paper addresses important issues in the study of miscarriages and abortion, especially as it relates to understanding miscarriage reporting, yet, the paper is confusingly framed and the analyses have important limitations that are not sufficiently justified or explained.

Reviewer #2: The utility of latent class analysis method, which is a powerful method for unobservable latent constructs, provides a new insight to classify respondents into “underreporter” and “reporter” groups. I think, main novelty of the study is to test utility of LCA method when estimating accurate abortion reports in a comparative way with survey reports, without any need to medical records. As you have stated in your paper, underreporting of abortions within the scope of mode differences is well-studied issue for NSFG data. However, constructing new scales on sexual attitudes and attitudes towards marriage contributes to current literature on determining factors behind underreporting behaviors of respondents.

So, I recommend revisions based on the following statements:

1- I suggest more specific manuscript title in accordance with study interests:

“Detecting Underreporters of Abortions and Miscarriages in the U.S.: A Latent Class Analysis Approach from NSFG, 2011-2015”.

2- Although the paper focuses on underreporting of abortions and miscarriages in the U.S., it is also a significant issue for other country settings. Moreover, various methodological approaches were tried to detect underreporters, and calculate accurate abortion rates at the end.

I suggest some recent works for background section. These studies include new techniques to understand underreporting for the U.S. and other countries, as well as characteristics of underreporters.

Lara, D., Strickler, J., Olavarrieta, C.D., & Ellertson, C. (2004). Measuring induced abortion in Mexico: a comparison of four methodologies. Sociological Methods & Research 32(4), 529–558.

Saraç, M., & Koç, İ. (2019). Increasing misreporting levels of induced abortion in Turkey: is this due to social desirability bias?. Journal of Biosocial Science, 52(2), 213-229.

Anderson, B.A., Katus, K., Puur, A. & Silver, B.D. (1994). The validity of survey responses on abortion: evidence from Estonia. Demography 31(1), 115–132

Tennekoon, V. S. (2017). Counting unreported abortions: A binomial-thinned zero-inflated Poisson model. Demographic Research, 36, 41-72.

Medeiros M & Diniz D (2012) Recommendations for abortion surveys using the ballot-box technique. Ciência & Saúde Coletiva 17(7), 1721–1724.

3- I suggest adding ‘abortion’, ‘miscarriage’, and ‘NSFG’ to the keywords.

4- Line 35: The authors stated that abortion is more stigmatized than miscarriage. There might be some cultural and political reasons behind that argument. The authors should provide justifications/works for that argument.

5- Line 64: ..... to understand why abortions are so badly underreported in surveys by..... Instead of that, I recommend ..... to understand factors affecting underreporting behavior of respondents by identifying the characteristics .... I think, this is more appropriate compared to “why” question, when the authors’ statistical analyses are considered.

6- Starting from Line 64: A bit difficult to follow study aims. The authors should re-organize study goals as the following. My suggestion is also more appropriate for the flow of sections about analytical strategy and results.

First goal: Evaluating utility of the LCA models in identifying underreporters of abortions/miscarriages.

Second goal: Investigating characteristics of underreporters of abortions/miscarriages.

Third goal: Determining common and differential features of respondents who underreport their abortions and miscarriages.

7- Line 124: For the readers, the authors should add a footnote for explanations of false positive and false negative errors.

8- Line 132: For a two-class LCA, three indicators are necessary for the model to be identification. The authors should add a reference here for required number of indicators to achieve model identification.

9- Line 173: … other variables. The authors should clarify on which stratification variables were used in the NSFG’s sample design, except for census division and population parameters?

10- Line 182: …. selecting one of the eligible persons for the main interview. The authors should clarify on which selection method were used to select an eligible person to interview? (Kish method, birthday method or else?)

11- Line 191: Author should give a brief information about sensitive questions in ACASI part of the questionnaire (except for abortion and miscarriage), to give a general picture for readers.

12- The authors defined “underreporter”, if a respondent who reported abortion/miscarriage in only one mode (regardless of its type; CAPI or ACASI). Instead, in accordance with the current literature, I am suggesting to define “underreporter”, if a respondent reported abortion in ACASI mode, but at the same time did not report it in CAPI mode.

As the authors discussed comprehensively in background section, respondents tend to underreport sensitive issues in face-to-face modes (i.e. CAPI) compared to self-administered modes (i.e. ACASI).

Replicating same analyses for that sub-group of respondents (ACASI reporter, CAPI underreporter) and re-organizing the paper would be useful. It is also more appropriate for the current literature and structural integrity of the paper.

13- Line 232: We used PROC LCA in SAS to fit the models.

Were the complex sample design features adjusted into LCA models?, likewise binary logistic regression analysis mentioned in Line 282. If yes, the authors should give information about that procedure, too. If no, I suggest incorporating weight, psu, and stratification variables into PROC LCA procedure in SAS.

14- Line 234: …these assignments were based on posterior class membership probabilities. This statement should be excluded. Because it has already stated within second method (Line 239).

15- Line 241: … we divided the respondent into four groups…

I suggest moving text into an organised scheme that is prepared to classify these sub-groups. It would be more useful for readers. In this scheme, the authors should provide unweighted case numbers for women groups.

16- Line 300: I guess, referring Table 2 should be replaced with Table 1 that includes survey reports.

17- Line 330: The two different methods of identifying underreporters of abortion produce very similar results. This result shows us validity/success of LCA models, authors should a sentence about that here.

18- Table 4 should include a note for representation of significance levels, as the authors added for Table 6.

19- In my opinion, results about live births should be excluded from Table 5 and scope of the paper. Abortions and miscarriages are quite relevant with literature and authors’ study objectives. There is no need to mention about live births under the results of miscarriages. Thus, I recommend restrict Table 5 only with miscarriages.

20- Line 374-377: Authors should add more discussions about the results on differential characteristics between underreporters of miscarriages and abortion. Why minority groups and marital status produce different results for two groups? Authors may refer evidence from literature here. It is also valid for discussion section (Line 428-434), too. Adding a brief explanation would be useful.

21- At the end of the discussion section, I would like to see three points:

-Can LCA method be an alternative way for other surveys in countries where there are no available medical records or, when there is only one mode of data collection?

-Could authors suggest any practical implications based on study results, especially when underreporters’ characteristics are considered? What can authors suggest for survey stages (e.g. questionnaire design, interviewer training, and data collection).

-To light future studies, which topics related to survey research are suggested by authors including use of LCA models?

22- Authors should provide exploratory factor analysis results to construct scales (like S1 Table and S2 table). The loadings of items under extracted factors and explained variances of factors should be provided in that table so that readers could follow results easily.

Reviewer #3: Broad reservation:

There is a tension in the article regarding its purpose.

The abstract and the paper look different. The abstract is focused on having two different measures of abortion and miscarriage and evaluating their consistency. This is interesting in itself and can be done focusing only on these two evaluations (and possibly analyzing with regression or means, who are the “inconsistent” reporters).

The paper is an application of latent class analysis that, in my opinion, is more debatable. Latent class analysis is about finding groups in the data, but these groups have no name. You do not get a class of “Had an abortion in last five years” vs “Did not have an abortion in last five years” as in table 2. That is pushing the method too much. In particular, when you need to introduce third unrelated variables for identification. If you bring marital status, as you do for instance, the two groups would be people who tend to have one of the marital statuses and report abortion or miscarriage in some of the dimensions and another that does not. This is different from the interpretation that I quoted. I think the paper is good while it develops the analysis in table 1, but the application of LCA is more troublesome and its interpretation should be much more cautious. I would redo or remove the LCA part.

You should think what is the purpose of the article. If you think what the abstract says is the relevant part, the article should not be as it is.

Other comments:

Based on the reservation stated above, and independently of where the article heads for, the abstract needs to be rewritten, perhaps a structured format will help. Currently, it does not talk about the results or implications of the study.

Missing recent references on underreporting in NSFG such as

Desai, S., Lindberg, L.D., Maddow-Zimet, I. et al. The Impact of Abortion Underreporting on Pregnancy Data and Related Research. Matern Child Health J 25, 1187–1192 (2021). https://doi.org/10.1007/s10995-021-03157-9

Suggestion to overcome the problems in identification:

The use of marital status as the 3rd variable makes the analysis weak since its direct association with abortion and miscarriage is unclear. Perhaps you could introduce a variable more connected to abortion and miscarriage. A possibility would be interbirth Interval (or the first birth interval for first births). The longer the last interbirth interval (or the ongoing birth interval), the more likely that missreporting is taking place. Having 3 variables measuring (somehow) the same phenomena could make the LCA interpretation more credible.

Other technical comments, although I believe the LCA and logit analysis have to be changed:

The logistic regression described in line 337 does not specify how variables have been selected. According to PLOS ONE statistical guidelines you should specify the procedure used for the identification of variables. The candidate variables should also be listed providing a rationale for the specification in the methods section. Also whether you checked for possible multicollinearity, and given the use of logistic regression, an assessment of balance. You could report it in an appendix table. It now seems that you are just putting together all those variables.

Reporting of the age-squared terms in table 4 (and 6) is wrong, showing only 0s. You should rescale so that the SE and the coefficients are shown, maybe multiplying them both by 1000. If they are all non-significant maybe it is best to remove them. Age-squared increases very much SEs given multicollinearity with age.

There is no overall statement of the probability of underreporting using weights. They could be added to tables 3 and 5.

Reviewer #4: Dear authors

I have read your manuscript with enthusiasm and enjoyed it. I have some minor comments and hope it improves the quality of the manuscript.

1. You noted that one of the aims is to understand why abortions are so badly underreported in surveys. I think it is better to say 'to quantify the level of underreporting'. I think qualitative-type works are needed to address WHY underreporting happens

2. What does term 'indicator variable' mean in LCA? Do you mean independent variables offered to the model so as to predict group membership? I guess you haven't dome that, because you did not want predictors of the group membership to be independent variables in the logistic regression models. I think it is worth to address these issues in the discussion.

3. My understanding is that LCA can provide better estimates if we include more variables in the model. Why only 2-indicator and 3-indicator variable models are developed? Wasn't is possible to include more variables in the LCA analysis?

4. LCA may classify some women to the over-report category. I liked the idea to force the model not to allocate any women to this category, and you showed that the results were robust. I am just curious how to interpret over-reported cases. Why some women should over-report abortion? Wasn't is possible to compare characteristics of truthful group with over-reported group?

5. The first paragraph in the results section can be deleted

6. How these data are calculated? "From 6.2% (n=703) to 6.6% (n=741) of the women were assigned to the “had an abortion” class, which is a bit higher than the proportion of self-reported abortions in either CAPI or ACASI modes.

7. Tables 2 and 3 show unweighted class memberships. What does unweighted mean here? Furthermore, in line 300, you noted that "weighted figures are similar". What does weighted data mean? Which weights and how applied?

8. The logistic regression outputs are not informative. I prefer to see OR, 95% CI, and P-value. Please explain on what basis variables were selected to be offered to the model? Why variable selection method, such as Backward Elimination, has not been applied? Also please report exact P-values instead of <0.10 or <0.05

6. PLOS authors have the option to publish the peer review history of their article (what does this mean?). If published, this will include your full peer review and any attached files.

Reviewer #1: No

Reviewer #2: No

Reviewer #3: **Yes: **José A. Ortega

Reviewer #4: **Yes: **Mohammad Reza Baneshi

---

## [Author Response · Author response to Decision Letter 0]

18 Apr 2022

Responses to specific reviewer and editor comments can be found in the uploaded "Rebuttal Letter". Thanks.

---

## [Decision Letter · Decision Letter 1]

25 May 2022

PONE-D-21-29890R1Detecting Underreporters of Abortions and Miscarriages in the National Study of Family Growth, 2011-2015PLOS ONE

Dear Dr. Yan,

Thank you for submitting your manuscript to PLOS ONE. After careful consideration, we feel that it has merit but does not fully meet PLOS ONE’s publication criteria as it currently stands. Therefore, we invite you to submit a revised version of the manuscript that addresses the points raised during the review process.

We look forward to receiving your revised manuscript.

Kind regards,

Janet E Rosenbaum, Ph.D.

Academic Editor

PLOS ONE

Journal Requirements:

Reviewers' comments:

Reviewer's Responses to Questions

**Comments to the Author**

1. If the authors have adequately addressed your comments raised in a previous round of review and you feel that this manuscript is now acceptable for publication, you may indicate that here to bypass the “Comments to the Author” section, enter your conflict of interest statement in the “Confidential to Editor” section, and submit your "Accept" recommendation.

Reviewer #1: (No Response)

Reviewer #2: (No Response)

2. Is the manuscript technically sound, and do the data support the conclusions?

Reviewer #1: Yes

Reviewer #2: Yes

3. Has the statistical analysis been performed appropriately and rigorously? 

Reviewer #1: Yes

Reviewer #2: Yes

4. Have the authors made all data underlying the findings in their manuscript fully available?

Reviewer #1: Yes

Reviewer #2: Yes

5. Is the manuscript presented in an intelligible fashion and written in standard English?

Reviewer #1: Yes

Reviewer #2: Yes

6. Review Comments to the Author

Reviewer #1: The revisions to the paper have greatly improved the clarity of the paper. I would suggest a few minor changes.

First, the comments from the authors on my second point in the first review were helpful, and the revisions begin to address this issue. However, I think it would be helpful in the discussion (around lines 435-436, where revisions were made) for the authors to either provide information or use their expertise to speculate about how similar/dissimilar the underreporters they are researching (i.e., those who report on only one CAPI or ACASI) might be to other kinds of underreporters. A sentence or two in the discussion would suffice.

Second, the clarifications provided to my comment regarding “truthful reporters” was also helpful. However, in line 158, the authors state that those who report on both CAP and ACASI are “truthful” reports of abortion, while those that report in only one are likely underreporters. It would be more accurate to call the former “consistent” reporters of abortion here. Alternatively, make explicit that you are going to assume that these consistent reporters are “truthful.” A sentence making that statement would suffice. The point about the LCA classifying “truthful” reporters is acceptable, but your argumentation should be stated clearly in the paper text. I suggest adding a brief explanation as provided in the reviewer comments in the analytic strategy section.

Other Minor issues:

Lines 45-47 [There is limited evidence… and Still…] sentences seem contradictory – maybe rephrase to clarify meaning.

Line 34 – the following citation should also be included re: Add Health: Tierney, Katherine I. 2019. “Abortion Underreporting in Add Health: Findings and Implications.” Population Research and Policy Review. doi: 10.1007/s11113-019-09511-8.

Reviewer #2: Comments to Authors:

The new version of the paper has become more understandable and clear for readers. I have just some minor comments for you, and these are related to my prior comments.

So, I recommend minor revisions:

1- The authors stated that abortion is more stigmatized than miscarriage.

Using the findings from the study [11] to justify this argument is better. Additionally, could you add some findings from the study [12], too?

The index that Bommaraju et al. (2016) used for abortion underreporters, shows more stigma on reporting an abortion as opposed to reporting a miscarriage.

2- I understood that the logistic regression models using survey data only examined characteristics of women who reported abortion/miscarriage in ACASI but not in CAPI.

But, in the text (see page 15) you said:

i. “… we fit logistic regression models to compare those who underreported their abortions (i.e. reporting group 2 above) to truthful respondents (i.e. reporting group 1).”

ii. I see group 2 above on the same page: 2) those who were assigned to the class but did not report an abortion in at least one mode.

According to the statement in ii, respondents who report their abortions/miscarriages in CAPI but not in ACASI are included in Group 2). In that case, still, the logistic models were run over women who reported abortion/miscarriage in ACASI but not in CAPI? You know, these are two different subgroups of underreporters. Could you clarify it, please? You can prefer to change your statement as follows:

“… we fit logistic regression models to compare those who underreported their abortions (reported in ACASI but not in CAPI) to truthful respondents (reported in both modes).”

3- From your point of view based on the findings, LCA may not be regarded as a success when it is compared to the method using survey data.

However, I believe that the LCA technique could be a considerable way to detect under-reporters of abortions/miscarriages, especially for different surveys that do not provide two data collection modes to gather the same information. Could you add something like that in your discussion, considering surveys that have different designs?

I believe that journal readers would like to use the LCA technique to detect underreporters of abortions/miscarriages using data coming from different surveys.

4- Lastly, the LCA method did not bring much advantage in identifying of underreporters of abortions in the NSFG.

However, do you suggest the use of this method when detecting underreporters of other sensitive variables in the NSFG (such as alcohol and drug use, involuntary sex and sexual disease?)

I think, the lack of advantage of the method for abortion/miscarriage underreporters may turn when underreporting of other variables are studied.

7. PLOS authors have the option to publish the peer review history of their article (what does this mean?). If published, this will include your full peer review and any attached files.

Reviewer #1: No

Reviewer #2: No

---

## [Author Response · Author response to Decision Letter 1]

8 Jun 2022

Dear Editor,

We have revised our manuscript to address comments from reviewers. Below we explained how we addressed each comment. We hope that this revision is satisfactory to you. 

Best Regard

Ting Yan and Roger Tourangeau

Reviewers’ comments: 

Reviewer #1: 

-First, the comments from the authors on my second point in the first review were helpful, and the revisions begin to address this issue. However, I think it would be helpful in the discussion (around lines 435-436, where revisions were made) for the authors to either provide information or use their expertise to speculate about how similar/dissimilar the underreporters they are researching (i.e., those who report on only one CAPI or ACASI) might be to other kinds of underreporters. A sentence or two in the discussion would suffice.

-We added our speculation on differences between women who denied reporting in both modes and women who reported in ACASI but not in CAPI I in the discussion (starting with line 441).

-Second, the clarifications provided to my comment regarding “truthful reporters” was also helpful. However, in line 158, the authors state that those who report on both CAP and ACASI are “truthful” reports of abortion, while those that report in only one are likely underreporters. It would be more accurate to call the former “consistent” reporters of abortion here. Alternatively, make explicit that you are going to assume that these consistent reporters are “truthful.” A sentence making that statement would suffice. The point about the LCA classifying “truthful” reporters is acceptable, but your argumentation should be stated clearly in the paper text. I suggest adding a brief explanation as provided in the reviewer comments in the analytic strategy section.

-In Line 158, we changed the word ‘truthful’ to ‘consistent’ and added a sentence at the end of that paragraph clearly stating that we are assuming consistent reporters are truthful reporters (starting with line 166). 

Other Minor issues:

Lines 45-47 [There is limited evidence… and Still…] sentences seem contradictory – maybe rephrase to clarify meaning.

-We rephrased the sentence to clarify meaning.

Line 34 – the following citation should also be included re: Add Health: Tierney, Katherine I. 2019. “Abortion Underreporting in Add Health: Findings and Implications.” Population Research and Policy Review. doi: 10.1007/s11113-019-09511-8.

-Thank you for the citation. We’ve added it.

Reviewer #2: 

1- The authors stated that abortion is more stigmatized than miscarriage.

Using the findings from the study [11] to justify this argument is better. Additionally, could you add some findings from the study [12], too?

The index that Bommaraju et al. (2016) used for abortion underreporters, shows more stigma on reporting an abortion as opposed to reporting a miscarriage.

-Thank you for the suggestion. We’ve added one finding reported in Table 3 of Bommaraju et al. (2016), starting with line 40 .

2- I understood that the logistic regression models using survey data only examined characteristics of women who reported abortion/miscarriage in ACASI but not in CAPI.

But, in the text (see page 15) you said:

i. “… we fit logistic regression models to compare those who underreported their abortions (i.e. reporting group 2 above) to truthful respondents (i.e. reporting group 1).”

ii. I see group 2 above on the same page: 2) those who were assigned to the class but did not report an abortion in at least one mode.

According to the statement in ii, respondents who report their abortions/miscarriages in CAPI but not in ACASI are included in Group 2). In that case, still, the logistic models were run over women who reported abortion/miscarriage in ACASI but not in CAPI? You know, these are two different subgroups of underreporters. Could you clarify it, please? You can prefer to change your statement as follows:

“… we fit logistic regression models to compare those who underreported their abortions (reported in ACASI but not in CAPI) to truthful respondents (reported in both modes).”

- The four reporting groups described on p10 are the results of comparing women’s actual answers to the latent class they were assigned to by LCA. Both Group 1 and Group 2 were assigned to the latent class of “having an abortion.” Group 1 reported having an abortion in both modes and, as a result, they are labeled as ‘truthful reporters of abortion’. By contrast, Group 2 respondents failed to report having an abortion in either CAPI or ACASI. As a result they are labeled as ‘underreporters of abortion’. 

We added some language to clarify the meaning of Group 2 (Lines 220-221, 224). 

3- From your point of view based on the findings, LCA may not be regarded as a success when it is compared to the method using survey data.

However, I believe that the LCA technique could be a considerable way to detect under-reporters of abortions/miscarriages, especially for different surveys that do not provide two data collection modes to gather the same information. Could you add something like that in your discussion, considering surveys that have different designs?

I believe that journal readers would like to use the LCA technique to detect underreporters of abortions/miscarriages using data coming from different surveys.

-We’ve added a paragraph about the use of LCA for other surveys in the discussion (p24, Lines 463-467). 

4- Lastly, the LCA method did not bring much advantage in identifying of underreporters of abortions in the NSFG.

However, do you suggest the use of this method when detecting underreporters of other sensitive variables in the NSFG (such as alcohol and drug use, involuntary sex and sexual disease?)

I think, the lack of advantage of the method for abortion/miscarriage underreporters may turn when underreporting of other variables are studied.

-We’ve added a couple sentence about the use of LCA for other variables in the discussion. (p24, Lines 463-467).

---

## [Decision Letter · Decision Letter 2]

28 Jun 2022

Detecting Underreporters of Abortions and Miscarriages in the National Study of Family Growth, 2011-2015

PONE-D-21-29890R2

Dear Dr. Yan,

We’re pleased to inform you that your manuscript has been judged scientifically suitable for publication and will be formally accepted for publication once it meets all outstanding technical requirements.

Kind regards,

Janet E Rosenbaum, Ph.D.

Academic Editor

PLOS ONE

Additional Editor Comments (optional):

Reviewers' comments:

Reviewer's Responses to Questions

**Comments to the Author**

1. If the authors have adequately addressed your comments raised in a previous round of review and you feel that this manuscript is now acceptable for publication, you may indicate that here to bypass the “Comments to the Author” section, enter your conflict of interest statement in the “Confidential to Editor” section, and submit your "Accept" recommendation.

Reviewer #1: All comments have been addressed

Reviewer #2: (No Response)

2. Is the manuscript technically sound, and do the data support the conclusions?

Reviewer #1: Yes

Reviewer #2: Yes

3. Has the statistical analysis been performed appropriately and rigorously? 

Reviewer #1: Yes

Reviewer #2: Yes

4. Have the authors made all data underlying the findings in their manuscript fully available?

Reviewer #1: Yes

Reviewer #2: Yes

5. Is the manuscript presented in an intelligible fashion and written in standard English?

Reviewer #1: Yes

Reviewer #2: Yes

6. Review Comments to the Author

Reviewer #1: (No Response)

Reviewer #2: (No Response)

7. PLOS authors have the option to publish the peer review history of their article (what does this mean?). If published, this will include your full peer review and any attached files.

Reviewer #1: No

Reviewer #2: No

---

## [Editor Report · Acceptance letter]

7 Jul 2022

PONE-D-21-29890R2 

Detecting Underreporters of Abortions and Miscarriages in the National Study of Family Growth, 2011-2015 

Dear Dr. Yan:

I'm pleased to inform you that your manuscript has been deemed suitable for publication in PLOS ONE. Congratulations! Your manuscript is now with our production department. 

Kind regards, 

on behalf of

Dr. Janet E Rosenbaum 

Academic Editor

PLOS ONE